# Pedunculopontine glutamatergic neurons control spike patterning in substantia nigra dopaminergic neurons

Daniel J Galtieri[1], Chad M Estep[1], David L Wokosin[1], Stephen Traynelis[2], D James Surmeier[1]*

[1]Department of Physiology, Feinberg School of Medicine, Northwestern University, Chicago, United States; [2]Department of Pharmacology, Emory University, Atlanta, United States

**Abstract** Burst spiking in substantia nigra pars compacta (SNc) dopaminergic neurons is a key signaling event in the circuitry controlling goal-directed behavior. It is widely believed that this spiking mode depends upon an interaction between synaptic activation of N-methyl-D-aspartate receptors (NMDARs) and intrinsic oscillatory mechanisms. However, the role of specific neural networks in burst generation has not been defined. To begin filling this gap, SNc glutamatergic synapses arising from pedunculopotine nucleus (PPN) neurons were characterized using optical and electrophysiological approaches. These synapses were localized exclusively on the soma and proximal dendrites, placing them in a good location to influence spike generation. Indeed, optogenetic stimulation of PPN axons reliably evoked spiking in SNc dopaminergic neurons. Moreover, burst stimulation of PPN axons was faithfully followed, even in the presence of NMDAR antagonists. Thus, PPN-evoked burst spiking of SNc dopaminergic neurons in vivo may not only be extrinsically triggered, but extrinsically patterned as well.
DOI: https://doi.org/10.7554/eLife.30352.001

*For correspondence:
j-surmeier@northwestern.edu

**Competing interests:** The authors declare that no competing interests exist.

## Introduction

SNc dopaminergic (DA) neurons play a central role in modulating goal directed actions and habits, which are under the control of the basal ganglia. As such, a great deal of effort has been devoted to understanding what governs the activity of SNc DA neurons. The spiking behavior of these cells can be placed into one of two broad categories – a single spiking, low-frequency (<10 Hz) mode and a multi-spike, higher frequency, 'burst' mode (*Grace and Bunney, 1984a*, *1984b*; *Hyland et al., 2002*). The single spiking mode is critical for maintaining ambient levels of DA in target structures, while the burst mode is thought to be a fundamental signal for action selection and reward-based learning (*Schultz, 2007*, *2016*; *Tsai et al., 2009*).

The single spiking mode of SNc neurons is generated by an intrinsic pacemaking mechanism involving the cooperation of a number of $Na^+$, $K^+$, and $Ca^{2+}$ ion channels (*Shepard and Bunney, 1988*; *Nedergaard et al., 1993*; *Chan et al., 2007*; *Puopolo et al., 2007*; *Guzman et al., 2009*; *Ding et al., 2011*; *Kimm et al., 2015*). This behavior is preserved in the absence of synaptic input, and is observed in both dissociated and ex vivo brain slice preparations (*Chan et al., 2007*; *Puopolo et al., 2007*; *Guzman et al., 2009*; *Ding et al., 2011*). In contrast, burst spiking is lost in preparations lacking functionally intact synaptic connectivity (*Shepard and Bunney, 1991*; *Johnson and Wu, 2004*; *Blythe et al., 2007*) and is disrupted in vivo by local application of glutamatergic receptor antagonists (*Grace and Bunney, 1984b*; *Charlety et al., 1991*; *Overton and Clark, 1992*; *Smith and Grace, 1992*; *Chergui et al., 1993*), demonstrating the necessity of synaptic activity for the production of these events.

Because local infusion of NMDAR antagonists in vivo reduced burst activity in anesthetized rodents (*Overton and Clark, 1992*; *Chergui et al., 1993*), subsequent studies have focused on the potential mechanisms by which NMDARs might generate naturally occurring bursts. Indeed, NMDARs can amplify intrinsic oscillatory activity and promote the transition from slow, single spike pacemaking to a burst pattern (*Wilson and Callaway, 2000*; *Kuznetsov et al., 2006*; *Deister et al., 2009*; *Kuznetsova et al., 2010*; *Ha and Kuznetsov, 2013*). In contrast, α-amino-3-hydroxy-5-methyl-4-isoxazolepropionic acid (AMPA) receptors (AMPARs), the other ionotropic glutamate receptor found at SNc glutamatergic synapses, appear unable to produce burst activity like that observed in vivo (*Shepard and Bunney, 1991*; *Johnson and Wu, 2004*; *Deister et al., 2009*). However, local electrical stimulation of glutamatergic axons can induce burst-like spiking in SNc DA neurons that is dependent upon AMPARs (*Georges and Aston-Jones, 2002*; *Blythe et al., 2007*).

One of the missing pieces in this story is an interrogation of specific glutamatergic inputs to SNc DA neurons. Rabies virus tracing studies have identified a handful of glutamatergic neurons that synapse on SNc DA neurons, including those in the subthalamic nucleus (STN), cerebral cortex, the superior colliculus and the pedunculopontine nucleus (PPN) (*Watabe-Uchida et al., 2012*). Of these, the PPN is of particular interest because of the strength of its projection to SNc and its connectivity with the rest of the basal ganglia (*Clarke et al., 1987*; *Lavoie and Parent, 1994a*, *1994b*; *Charara et al., 1996*; *Mena-Segovia et al., 2004*; *Martinez-Gonzalez et al., 2011*). Interestingly, PPN neurons respond to environmental events in a way that resembles SNc DA neurons, including burst spiking in response to salient and rewarding stimuli, but at latencies shorter than those observed in DA neurons (*Condé et al., 1998*; *Kobayashi et al., 2002*; *Pan and Hyland, 2005*; *Norton et al., 2011*; *Thompson and Felsen, 2013*; *Hong and Hikosaka, 2014*). Moreover, activation of the PPN in vivo can evoke repetitive spiking in SNc DA neurons (*Scarnati et al., 1984*; *Lokwan et al., 1999*; *Floresco et al., 2003*; *Hong and Hikosaka, 2014*), and ablation of the PPN disrupts learning operant tasks (*Inglis et al., 2000*; *Wilson et al., 2009*; *Bortolanza et al., 2010*; *Syed et al., 2016*).

Taken together, these observations suggest that PPN glutamatergic neurons might be able to generate patterned activity in SNc DA neurons, particularly bursts. To isolate the influence of PPN glutamatergic neurons on SNc, a combination of optogenetic and pharmacological tools were used. Subcellular optogenetic mapping revealed that PPN glutamatergic synapses were focused on the soma and proximal dendrites, near the axon initial segment. Indeed, stimulation of PPN axons reliably evoked spikes within SNc neurons at a variety of firing rates, including those observed in in vivo bursts. The ability of PPN axons to drive spiking was dependent solely upon AMPARs, not NMDARs. Thus, in addition to NMDAR-dependent forms of burst spiking, where the pattern of activity is dependent upon an interaction between synaptic and intrinsic mechanisms, the pattern of PPN-evoked burst spiking may be extrinsically determined.

## Results

### PPN-SNc glutamatergic synapses had Ca$^{2+}$ impermeable AMPARs and GluN2D containing NMDARs

Injections of adeno-associated virus serotype 9 (AAV9) ChR2-eYFP driven by the human Synapsin I (hSyn) promoter were made in to the PPN of wild-type or DAT-Cre/Ai14-tdTomato mice (*Figure 1A*). Projections from the PPN to SNc were visualized ten days after injection (*Figure 1B–C*). To confirm functional expression of ChR2 in PPN neurons, whole-cell patch clamp recordings were performed on ChR2 expressing cells in the PPN (*Figure 1—figure supplement 1A*). Photo-stimulation of these cells with full-field (~360 μm diameter) blue light (473 nm) LED illumination reliably evoked photocurrents and associated action potentials (*Figure 1—figure supplement 1B–C*).

SNc neurons were identified in coronal or parasagittal ex vivo brain slices based on their morphology, location within the slice, and regular pacemaking activity (*Lacey et al., 1989*; *Mercuri et al., 1994*; *Chan et al., 2007*; *Guzman et al., 2009*). Glutamatergic responses were isolated by antagonizing GABA$_A$ (10 μM gabazine) and nicotinic cholinergic receptors (10 μM mecamylamine). Synaptic responses evoked with 1 ms full-field photo-stimulation of PPN ChR2 expressing afferents were recorded in SNc neurons held at −70 mV in the whole-cell patch-clamp configuration (*Figure 1D*). Paired-pulse stimulation (20 Hz) evoked responses with amplitude ratios less than one (*Figure 1E*;

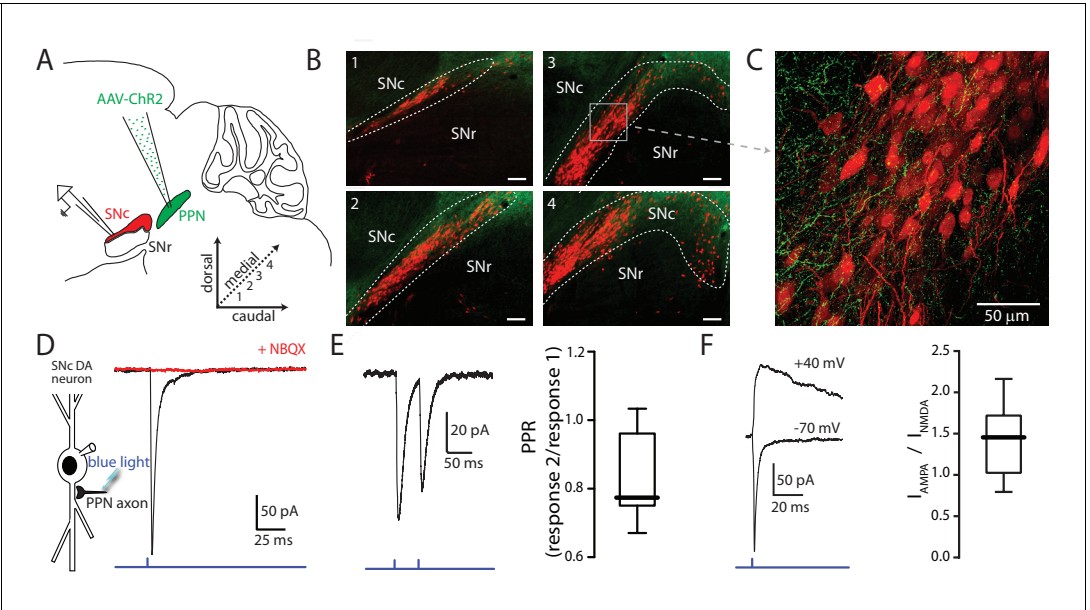

**Figure 1.** Optogenetic stimulation of PPN afferents evokes glutamatergic responses in SNc DA neurons. (**A**) Stereotaxic injections of AAV9.hSyn.hChR2 were made in to the PPN of DAT-Cre/Ai14-tdTomato or wild-type mice. (**B**) Representative images taken from four sagittal sections from one DAT-Cre/Ai14-tdTomato mouse 10 days following injection of hSyn-ChR2-eYFP in to PPN. Scale bars represent 100 μm. (**C**) Expanded view of section (gray box) from (B3) showing ChR2-expressing PPN fibers (green) intermingled with SNc DA neurons (red). (**D**) Blue light LED stimulation produced an inward current in SNc DA neurons held at −70 mV that was abolished by the AMPAR antagonist NBQX (10 μM). (**E**) A 20 Hz paired-pulse stimulation protocol produced a depressing synaptic response in SNc DA neurons. (Right) Summary of PPR responses (0.83 ± 0.13, n = 10). (**F**) To determine AMPA:NMDA ratio cells were initially held at −70 mV to determine the timing of the AMPA peak (bottom trace), after which cells were depolarized to + 40 mV to relieve $Mg^{2+}$ block of the NMDA receptor. The NMDA peak was calculated 40 ms after the AMPA peak. (Right) Summary of AMPA peak current vs. NMDA peak current (1.43 ± 0.42, n = 13).

DOI: https://doi.org/10.7554/eLife.30352.002

The following source data and figure supplements are available for figure 1:

**Source data 1.** Raw data points for panels E and F.
DOI: https://doi.org/10.7554/eLife.30352.006

**Figure supplement 1.** ChR2 is functionally expressed in PPN neurons.
DOI: https://doi.org/10.7554/eLife.30352.003

**Figure supplement 2.** Electrical stimulation of PPN glutamatergic afferents to SNc.
DOI: https://doi.org/10.7554/eLife.30352.004

**Figure supplement 2—source data 1.** Raw data points for summary plots.
DOI: https://doi.org/10.7554/eLife.30352.005

PPR = 0.83 ± 0.13, n = 10), indicating that the PPN glutamatergic synapse had a high release probability. This was confirmed with electrical stimulation of PPN afferents using a bipolar electrode placed in the rostral portion of PPN (*Figure 1—figure supplement 2*; PPR = 0.70 ± 0.16, n = 19). To determine the AMPA/NMDA ratio at these synapses, cells were first held at −70 mV to determine the time of the AMPA peak, and then held at + 40 mV to relieve NMDAR $Mg^{2+}$ block. The NMDA current was measured 40 ms after the AMPA peak (*Figure 1F*). Measured in this way, the NMDAR current was roughly half that of the AMPAR current (peak AMPAR current/peak NMDAR current = 1.43 ± 0.42, n = 13).

Next, the receptor subunit composition of the AMPARs and NMDARs was assessed. The majority of AMPARs in the brain contain edited GluA2 subunits, making them essentially impermeable to $Ca^{2+}$ (*Kawahara et al., 2003*; *Wright and Vissel, 2012*; *Henley and Wilkinson, 2016*). In contrast to AMPARs lacking edited GluA2 subunits, these AMPARs do not rectify at depolarized membrane potentials (*Koike et al., 1997*; *Washburn et al., 1997*). At PPN synapses on SNc dopaminergic neurons, there was no discernible rectification at depolarized membrane potentials (*Figure 2A*), suggesting AMPARs at this synapse had edited GluA2 subunits. Furthermore, the polyamine toxin philanthotoxin-74, which preferentially antagonizes GluA2-lacking (GluA1 and GluA3 homomeric)

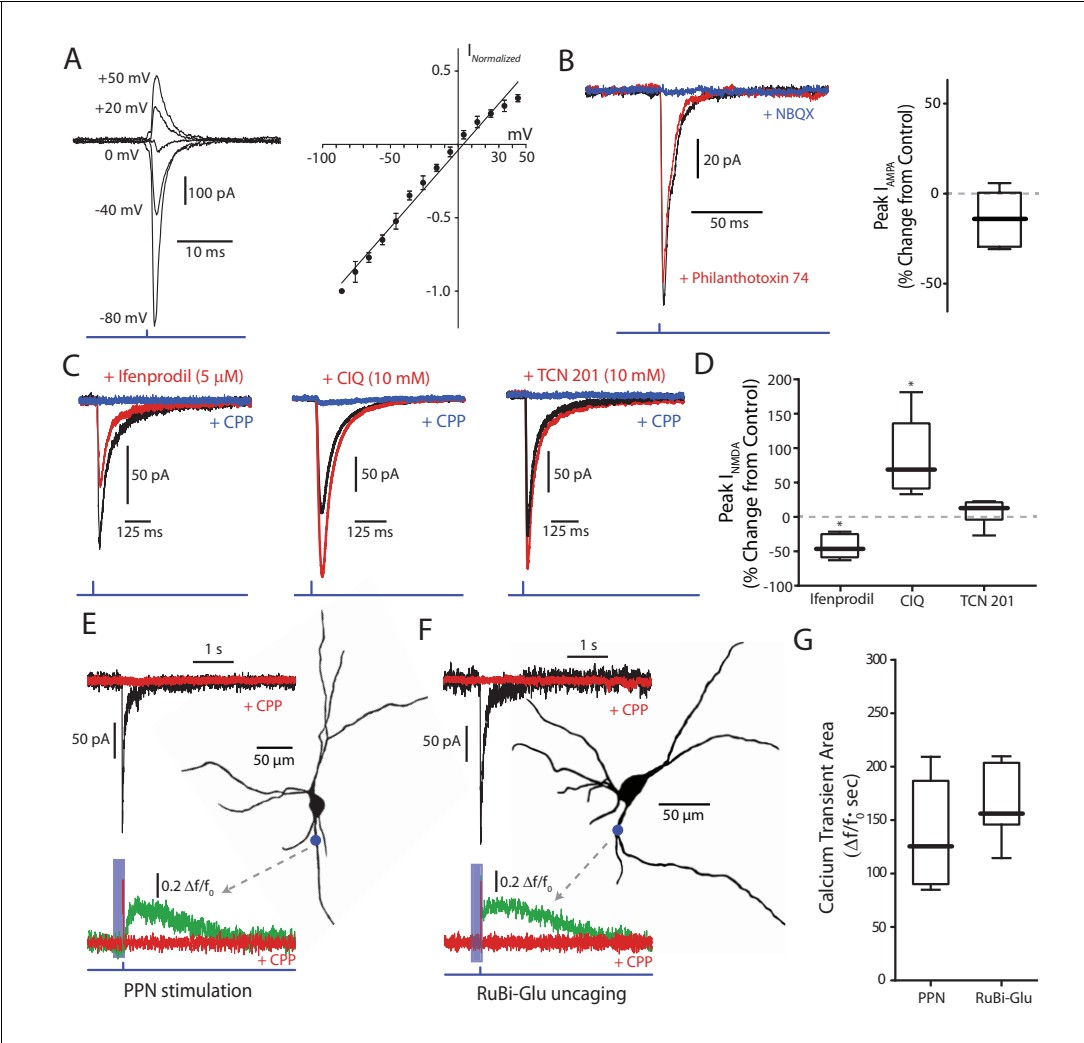

**Figure 2.** Glutamatergic receptors at the PPN-SNc synapse are composed of GluA2-containing AMPARs and GluN2B/D-containing NMDARs. (A) Example traces of isolated AMPA currents evoked with the cell held at different membrane potentials. (Right) Summary current-voltage (IV) relationship, where the currents at the different membrane potentials for each cell have been normalized to the current measured at −80 mV for that cell. A linear fit was applied to the data points (represented as mean ± standard deviation). No obvious rectification is observed at positive membrane potentials. (B) Example traces of AMPA currents recorded from a DA neuron held at −70 mV before and after pharmacological manipulation. Application of philanthotoxin-74 (5 μM; red) did not attenuate the measured current as compared to control (black). NBQX (10 μM; blue) was applied to confirm the identity of the current. (Right) Summary of the effect of philanthotoxin-74 application (−13.17% ± 14.67%, n = 6). Summary data represented as mean ± standard deviation. No significant effect was observed (p=0.1563). (C) Example traces of isolated NMDA currents measured from DA neurons held at −60 mV in aCSF containing 0 $Mg^{2+}$ before and after application of receptor-subunit specific pharmacological agents. (Left) The Glu2NB-selective antagonist ifenprodil attenuated the observed NMDA current, while (Middle) the Glu2NC/D-selective potentiator CIQ increased the peak of the measured current. (Right) The GluN2A-selective antagonist TCN 201 failed to attenuate the measured current. In all examples the nonspecific NMDAR antagonist CPP (5 μM) abolished the current, confirming its identity. (D) Summary of the effect, relative to control, on isolated NMDA currents of the GluN2B-selective antagonist ifenprodil (−43.31 ± 16.89%, n = 6), the GluN2C/D-selective potentiator CIQ (86.22 ± 56.54%, n = 6), and the GluN2A selective antagonist TCN201 (7.76 ± 18.40%, n = 6). Summarized data shown as mean ± standard deviation. Statistical tests were two-tailed Wilcoxon signed rank tests. *p<0.05. (E) 2PLSM calcium imaging of a PPN-evoked NMDA current and the associated calcium transient produced by that current. (Top) Example trace of a somatically recorded NMDA current produced by blue laser stimulation of a region of dendrite, visualized after filling the cell with a red indicator dye. (Right) Reconstructed cell generated, with a blue dot indicating both the point of stimulation and the general region where imaging data was acquired. (Bottom) Normalized (Δf/f₀) calcium transient associated with somatic NMDA current. Shaded blue area represents region where green PMT was shuttered. Application of CPP (5 μM; red) abolished both the somatically recorded NMDA current (top) and the associated calcium transient (bottom). (F) Similar to (E), with the primary difference being that the evoked response is produced by the uncaging of focally applied RuBi-glutamate (2 mM) by blue laser light. (G) Summary of calcium transient areas from PPN-evoked (139.26 ± 56.40Δf/f₀ * sec, n = 5) and uncaged RuBi-glutamate mediated (165.94 ± 40.27Δf/f₀ * sec, n = 5) NMDA currents. No significance difference was found between the two data sets (p=0.4206). Summary data presented as mean ± standard deviation. Statistical test used was a two-tailed Mann-Whitney test.

*Figure 2 continued on next page*

*Figure 2 continued*

DOI: https://doi.org/10.7554/eLife.30352.007

The following source data is available for figure 2:

**Source data 1.** Raw data points for panels A, B, D, and G.

DOI: https://doi.org/10.7554/eLife.30352.008

AMPARs (*Poulsen et al., 2014*), failed to significantly reduce the AMPAR currents (*Figure 2B*), again indicating the presence of the GluA2 receptor subunit.

NMDARs are obligate heteromers composed of two GluN1 and two GluN2 subunits. The identity of the two GluN2 subunits significantly impacts properties of the receptor, such as sensitivity to $Mg^{2+}$ block and relative $Ca^{2+}$ permeability. While the majority of NMDARs in the brain contain a combination of GluN2A and GluN2B subunits (*Monyer et al., 1994*; *Wyllie et al., 2013*), previous work has shown that NMDARs in SNc DA neurons contain GluN2B and GluN2D subunits (*Jones and Gibb, 2005*; *Brothwell et al., 2008*; *Suárez et al., 2010*; *Huang and Gibb, 2014*), with the latter conferring reduced $Mg^{2+}$ sensitivity and $Ca^{2+}$ permeability (*Siegler Retchless et al., 2012*; *Huang and Gibb, 2014*). However, none of this work was done at identified synapses. To determine the subunit composition of the NMDARs at PPN synapses, currents were evoked by a single 1 ms full-field LED pulse with SNc cells held at −60 mV in the nominal absence of extracellular $Mg^{2+}$. In agreement with previous work, both the GluN2B specific antagonist ifenprodil (*Williams, 1993*) and the GluN2C/D specific potentiator CIQ (*Mullasseril et al., 2010*) altered the amplitude of the evoked NMDAR currents (*Figure 2C–D*; ifenprodil % change: −43.31 ± 16.89, n = 6, p=0.0313; CIQ % change: 86.22 ± 56.54, n = 6, p=0.0313). In contrast, the GluN2A specific antagonist TCN 201 (*Edman et al., 2012*) had no effect on NMDAR currents (*Figure 2C–D*; % change: 7.76 ± 18.40, n = 6, p=0.4375).

To determine whether $Ca^{2+}$ entry through synaptic NMDARs differed from those that could be activated at neighboring dendritic regions, two-photon-excitation laser-scanning microscopy was used to image SNc dopaminergic neurons loaded with either 100 µm Fura-2 or Fluo-4 calcium indicator dye. The fluorescent transient evoked by focal, dendritic optogenetic stimulation of PPN axons was compared to the transient evoked by uncaging of RuBi-glutamate in the same dendritic region (*Figure 2E,F*). The $Ca^{2+}$ signal evoked by these two stimuli were not significantly different (*Figure 2G*).

## PPN glutamatergic synapses preferentially targeted proximal dendrites

To determine the dendritic location of PPN synapses, the sCRACM approach was used (*Petreanu et al., 2009*; *Fieblinger et al., 2014*). Briefly, in ex vivo brain slices from mice in which ChR2 was expressed in PPN and where conducted activity was blocked by tetrodotoxin (1 µM), a focused laser (473 nm) spot was moved in 8–10 µm increments along dendrites while using the somatic electrode to monitor for synaptically evoked currents (*Figure 3A–C*). Neurons with dendrites above or below the focal plane were excluded from study to minimize the chances that synaptically evoked activity originated from a site other than the one visualized. In these experiments, photo-stimulation of proximal dendrites (within ~70 um of the soma) reliably evoked responses from PPN ChR2-expressing terminals, whereas photo-stimulation of more distal dendrites almost always failed to evoke a response (*Figure 3A,E*).

Two additional experiments were performed to provide positive controls. First, glutamatergic synapses formed by STN neurons were mapped using the sCRACMapproach. In contrast to PPN synapses, STN synapses were found in both proximal and distal dendrites (*Figure 3C,E*). Next, RuBi-glutamate was uncaged (single-photon photolysis) along the dendrite using a focused laser (473 nm) spot. Robust responses were evoked in both proximal and distal dendrites (*Figure 3D,E*), arguing that there was a relatively uniform distribution of glutamatergic synapses throughout the SNc dendritic tree, in agreement with previous anatomical work (*Henny et al., 2012*). Thus, the apparent preferential localization of PPN synapses on proximal dendrites does not reflect a limitation of the sCRACm approach or the inability to detect activation of glutamatergic synapses at distal dendritic locations.

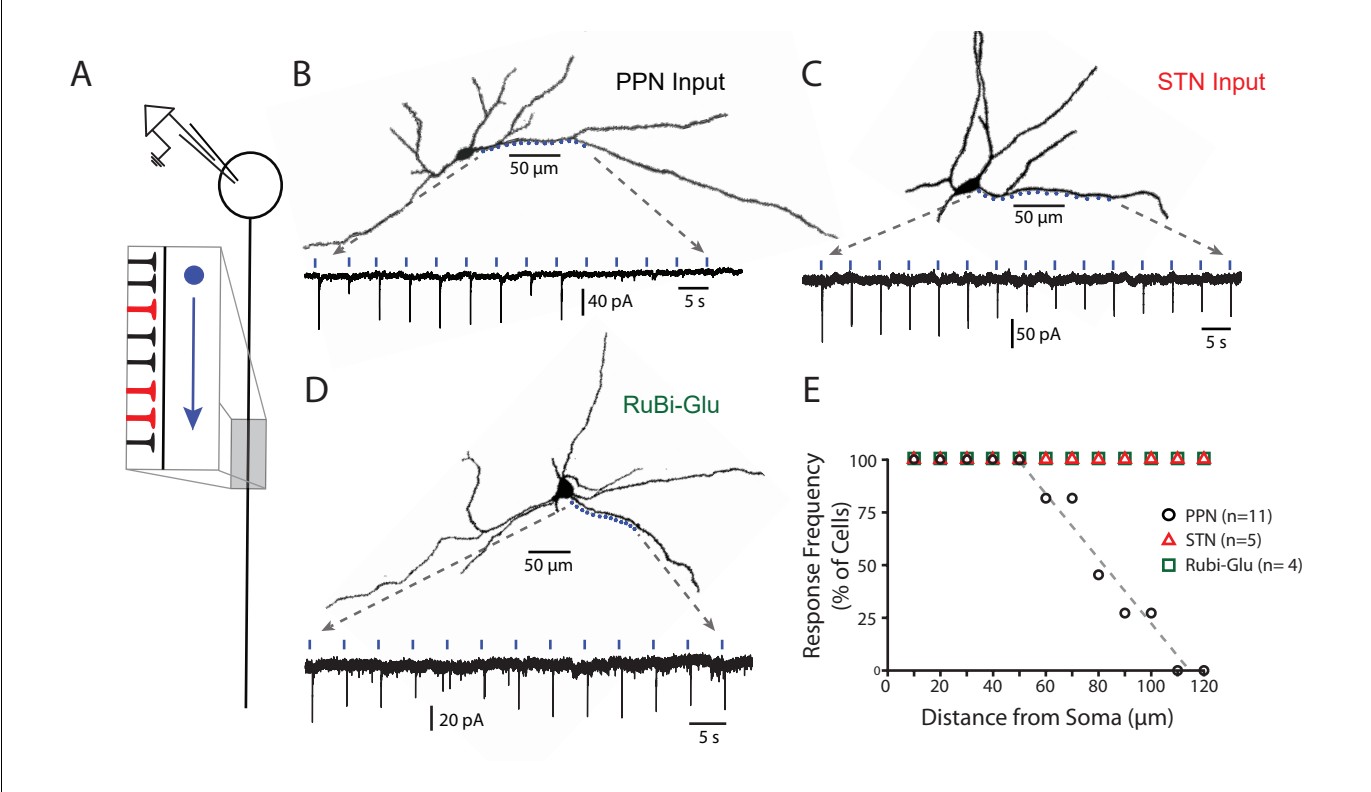

**Figure 3.** PPN glutamatergic synapses preferentially target proximal portions of the SNc dendritic tree. (**A**) Schematic of the experimental procedure. Spots along a section of dendrite, visualized with 2PLSM imaging of dye-filled cells, were assessed for responses to focal spot-laser stimulation using the sCRACM technique. (**B**) Example reconstructed SNc DA neuron (top) with stimulation spots placed in approximately 10 µm along a section of dendrite that was wholly visualized in the same focal plane. (Bottom) Somatically recorded currents produced by focal stimulation of PPN synapses at the spots displayed above. Responses were elicited in the first 9 spots (~0 to 80 µm from soma), while more distal spots (>100 µm) failed to elicit responses. (**C**) Similar to (**B**), with the primary being that the afferents expressing ChR2 in this example originated from the STN, rather than the PPN. In contrast to (**B**) responses were reliably elicited at both proximal and distal locations. (**D**) Similar to (**B**) and (**C**), with the primary difference being that evoked responses were produced by uncaging of focally applied RuBi-glutamate (2 mM). Similar to (**C**), and in contrast to (**B**), response were elicited at in both proximal and distal dendritic regions. (**E**) Summary response frequency at points along the dendrite, measured as the fraction of cells that showed a synaptic response at a particular distance from the soma. Responses recorded from DA neurons in slices expressing ChR2 in PPN afferents (n = 11) showed a location-dependent decrement in response frequency that was not observed in STN-ChR2 evoked responses (n = 5) or RuBi-glutamate mediated responses (n = 4).

DOI: https://doi.org/10.7554/eLife.30352.009

The following source data and figure supplements are available for figure 3:

**Source data 1.** Raw data points for summary plot.
DOI: https://doi.org/10.7554/eLife.30352.012

**Figure supplement 1.** Validation of the spatial resolution for sCRACM functional mapping.
DOI: https://doi.org/10.7554/eLife.30352.010

**Figure supplement 1—source data 1.** Raw data points for summary plot.
DOI: https://doi.org/10.7554/eLife.30352.011

## PPN glutamatergic synapses were capable of spike patterning

Previous work has shown that proximal glutamatergic synapses onto SNc dopaminergic neurons are better able to drive spiking than more distal ones (*Blythe et al., 2009*). To determine the ability of proximal PPN synapses to drive patterned spiking, SNc dopaminergic neurons were recorded in the perforated patch mode and PPN axons stimulated optogenetically. The probability of evoking a spike in an SNc DA neuron rose rapidly with stimulus intensity (*Figure 4—figure supplement 1*). Using the lowest LED power that reliably evoked a spike (typically 6–10% of the LED maximum power), stimulus trains of varying length and frequency were delivered. In addition to regular trains, burst patterns similar to those recorded from SNc dopaminergic neurons in vivo were included

(*Grace and Bunney, 1984b*). To ensure response fidelity for the higher-frequency (>10 Hz) stimulation protocols, the opsin Chronos was used in a subset of these experiments (*Klapoetke et al., 2014*). Regardless of the protocol used, optical stimulation of PPN axons reliably evoked spikes throughout the stimulus train (*Figure 4A–B*), with the average spike frequency within the stimulus period being linearly related (with a slope of 1) to the stimulus frequency (*Figure 4C*).

To determine whether NMDARs contributed to the ability of PPN axons to drive burst spiking, a pharmacological approach was used. Surprisingly, application of the NMDAR antagonist CPP, did not significantly alter the response to PPN stimulation at any frequency (*Figure 4D–F*). In contrast, application of the AMPAR antagonist NBQX completely abolished responses (*Figure 4D–F*), indicating a dependence on AMPARs, but not NMDARs.

Frequently, PPN-evoked spikes were truncated in amplitude and had a more hyperpolarized threshold than spikes that were spontaneously generated (e.g., *Figure 4B*). These likely represent axon initial segment (AIS) spikes that failed to invade the somato-dendritic (SD) region where the electrode was positioned (*Grace and Bunney, 1980*, *1983a*, *1983b*). Indeed, previous work has shown that antidromically-evoked AIS spikes often fail to generate full SD spikes in SNc DA neurons (*Grace and Bunney, 1983b*). The ability of AIS spikes to evoked full SD spikes should be dependent upon the excitability of the SD region, which should be determined by ongoing pacemaking. To test this hypothesis, individual spikes were evoked by optical stimulation of PPN axons at different points in the pacemaking cycle (*Figure 5A*). When the stimulus occurred towards the end of normal pacemaking cycle, the evoked spikes were indistinguishable from spontaneously occurring spikes (*Figure 5A–B*). In contrast, when spikes were evoked early in the oscillation, just after a spontaneously occurring spike, they exhibit more hyperpolarized spike thresholds and reduced amplitudes, as evident in plots of the first derivative of membrane voltage (dV/dt) as a function of membrane voltage (mV) (*Figure 5A,C*). Plots of PPN-evoked spike threshold and amplitude as a function of the time from the preceding spike, normalized by the average interspike-interval (ISI) of the spikes preceding the stimulus (4 s worth of recording) revealed this relationship more clearly (*Figure 5D–E*). PPN-evoked spikes early in pacemaking cycle (i.e. near time = 0) had hyperpolarized thresholds and reduced amplitudes, whereas spikes evoked near the end of the pacemaking cycle were identical to spontaneously occurring spikes. Also, in agreement with previous work (*Guzman et al., 2009*), PPN-evoked spikes reset the pacemaking cycle, as indicated by a clustering of points around one when comparing the ISI between the evoked spike and the next spontaneously occurring spike to the mean ISI for the cell (*Figure 5F*).

Although PPN-evoked spikes were not strongly influenced by somatic conductances, they should be regulated by dendritic conductances because of the common dendritic location of the AIS (*Blythe et al., 2009*; *Matsuda et al., 2009*). One well described dendritic channel that slows repetitive spiking in SNc DA neurons is the small conductance, $Ca^{2+}$-activated $K^+$ channel (SK) (*Ping and Shepard, 1996*; *Wolfart et al., 2001*). Indeed, in addition to accelerating pacemaking rate, blocking SK channels with apamin increased the ability of PPN terminals expressing ChR2 to evoke faithful, repetitive (10 Hz) spiking at low stimulus intensities (*Figure 5—figure supplement 1*). Interestingly, PPN stimulation in the presence of apamin delayed the next spontaneously occurring spike, rather than simply resetting the pacemaking rhythm (*Figure 5—figure supplement 1*).

## PPN-evoked spikes propagated down the axon

As mentioned above, the AIS typically arises from a proximal dendrite in SNc DA neurons (*Blythe et al., 2009*; *Matsuda et al., 2009*), in the region where PPN synapses were found. Thus, it was possible that PPN-evoked spikes, even though they often appeared truncated at the soma, would be faithfully propagated down the axon. To test this hypothesis, paired recordings were performed from the soma and from the axon. First, a somatic whole cell recording was established and the cell filled with dye to allow visualization of the dendrites and axon. The axon was identified by the presence of a retraction ball (*Atherton et al., 2008*; *Blythe et al., 2009*). Once identified, a second pipette was used to record from the axon in a loose-seal configuration. Recordings were then simultaneously made of both spontaneous and evoked spikes from the soma and axon (*Figure 6B–C*). Spontaneously recorded somatic spikes invariably propagated into the axon, as expected. More importantly, PPN-evoked spikes were also invariably seen in the axon, regardless of the phase of the pacemaking cycle and the somatic appearance of the spike (*Figure 6B–C*). Plotting the instantaneous spike frequency within the axon as a function of the instantaneous spike frequency in the

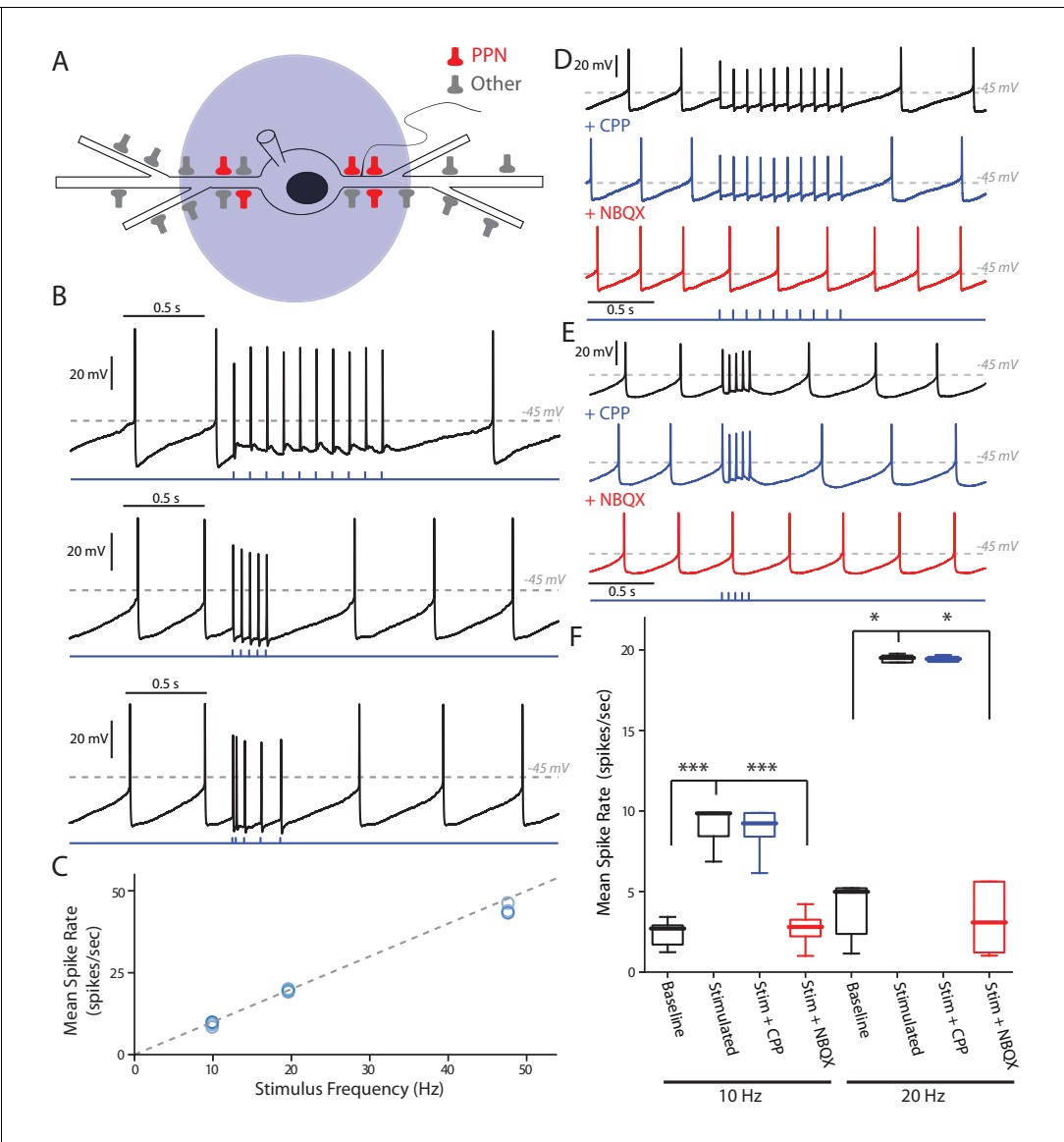

**Figure 4.** PPN glutamatergic input is capable of patterning SNc DA neuron firing. (**A**) Full field blue-LED stimulation of ChR2-expressing PPN afferents was performed while recording spiking activity in SNc DA neurons using the perforated-patch configuration. (**B**) Example traces of different stimulation patterns: (Top) 10, 1 ms stimuli with an inter-stimulus interval of 100 ms, (Middle) 5, 1 ms stimuli with an inter-stimulus interval of 50 ms, and (Bottom) 5, 1 ms stimuli with inter-stimulus intervals of 20 ms, 50 ms, 100 ms, and 12 0 ms. In all cases spikes were reliably generated by the stimulation protocols. (**C**) Summary of the mean intra-stimulus spike frequency (spikes/s) as a function of the frequency of stimulation. The measured spike frequency was linearly related to the stimulation frequency in a near 1:1 relationship. (**D**) Example traces from a SNc DA neuron responding to the 10 stimuli, 10 Hz PPN ChR2 stimulation protocol before and after pharmacological manipulation. Application of the NMDAR antagonist, CPP (5 μM), did not significantly attenuate the response to PPN stimulation, while application of the AMPA receptor antagonist, NBQX (10 μM) completely abolished the evoked response. (**E**) Similar to (**D**), with the primary difference being the usage of the 5 stimuli, 20 Hz PPN stimulation protocol. As with (**D**), application of CPP failed to attenuate the response, while NBQX completely abolished the response. (**F**) Summary of pharmacological manipulation of SNc DA neuron firing pattern response to PPN stimulation. Both 10 Hz (9.03 ± 1.19 spikes/s, n = 9) and 20 Hz (19.53 ± 0.25 spikes/s, n = 5) stimulation significantly increased firing rate in comparison to control (10 Hz stim: 2.40 ± 0.82 spikes/s, n = 9, p=0.0001; 20 Hz stim: 3.43 ± 1.90 spikes/s, n = 5, p=0.0117). CPP application failed to significantly decrease the response to stimulation (10 Hz stim: 8.81 ± 1.55 spikes/s, n = 5, p=0.8413; 20 Hz stim: 19.49 ± 0.19 spikes/s, n = 5, p=1.0). In contrast, NBQX application significantly attenuated the PPN evoked response (10 Hz stim: 2.71 ± 1.10 spikes/s, n = 6, p=0.0008; 20 Hz stim: 2.73 ± 2.25 spikes/s, n = 5, p=0.0234). Summaries are presented as mean ± standard deviation. Statistical tests used were two-tailed Mann-Whitney tests with Holm-Bonferroni corrections for multiple comparisons. ***p<0.001, *p<0.05.

DOI: https://doi.org/10.7554/eLife.30352.013

The following source data and figure supplements are available for figure 4:

**Source data 1.** Raw data points for panels C and F.

*Figure 4 continued*

DOI: https://doi.org/10.7554/eLife.30352.016

**Figure supplement 1.** Evoked-spike probability is related to stimulus intensity.

DOI: https://doi.org/10.7554/eLife.30352.014

**Figure supplement 1—source data 1.** Raw data points for panel B.

DOI: https://doi.org/10.7554/eLife.30352.015

soma confirmed this relationship (*Figure 6D–E*). Failure of the somatically recorded spikes to invade the axon would lead to points below the linear trend line, but this was not seen as all somatic spikes showed corresponding axonal spikes (*Figure 6D–E*). This finding is in agreement with previous work showing that AIS spikes alone are capable of triggering axonal firing (*Grace and Bunney, 1983b*).

## Discussion

Burst spiking in SNc DA neurons is a critical signal for goal-directed behavior (*Schultz, 2007*; *Tsai et al., 2009*; *Bromberg-Martin et al., 2010*; *Schultz, 2016*). While dependent upon synaptic activity (*Grace and Bunney, 1984b*; *Overton and Clark, 1992*; *Smith and Grace, 1992*), the cellular mechanisms dictating the temporal structure of the burst are poorly understood. The most widely held view is that bursts are produced by intrinsic oscillatory mechanisms engaged by activation of dendritic NMDARs (*Esposito et al., 2009*; *Morikawa and Paladini, 2011*; *Paladini and Roeper, 2014*). Our results expand this landscape to include an additional mechanism by which bursts can be generated. In particular, they show that bursts could be produced by patterned stimulation of PPN synapses formed on proximal dendrites near the AIS. Engagement of NMDARs was not necessary for burst generation in this case. Moreover, in contrast to the conventional model, this mechanism allows bursts to be generated and structured independently of ongoing pacemaking activity or synaptic input to more distal dendrites, like that arising from tonic activity in pallidal and nigral GABAergic neurons. Being able to precisely control the timing and duration of bursts could prove to be important to movement control, particularly that triggered by external events.

### PPN glutamatergic synapses target proximal dendrites

Our experiments provide the first characterization of the receptor complement at PPN glutamatergic and the sub-cellular distribution of these synapses on SNc DA neurons. Optogenetic approaches allowed the selective activation of axons originating in the PPN. Pharmacological tools allowed the receptor subtypes at these synapses to be determined. As in most adult glutamatergic synapses, the AMPARs at PPN synapses were $Ca^{2+}$-impermeable, as judged by their lack of rectification and insensitivity to philanthotoxin-74. Moreover, as expected from previous work examining NMDARs at unidentified synapses (*Jones and Gibb, 2005*; *Brothwell et al., 2008*; *Suárez et al., 2010*), the NMDARs at PPN synapses were GluN2B/D containing. More specifically, the magnitude of the NMDAR block achieved by ifenprodil was that expected of a triheteromeric, GluN2B/D containing receptor (*Hatton and Paoletti, 2005*; *Huang and Gibb, 2014*). The ability of the GluN2C/D potentiator (CIQ) to increase NMDAR currents further supports the proposition that triheteromic GluNB/D containing receptors are present at this synapse (*Jones and Gibb, 2005*; *Huang and Gibb, 2014*).

Although the composition of ionotropic glutamate receptors at PPN synapses was expected, their sub-cellular distribution, as revealed by the sCRACM technique (*Petreanu et al., 2009*; *Fieblinger et al., 2014*), was not. In contrast to the broad distribution of postsynaptic glutamate receptors and STN synapses, PPN glutamatergic synapses were found only on proximal dendrites. This location specificity places these synapses near the AIS, which typically arises from proximal portions of a primary dendrite (*Blythe et al., 2009*; *Matsuda et al., 2009*). It remains to be determined whether synapses made by other glutamatergic regions innervating the SNc (e.g., superior colliculus) have a similar distribution.

### An alternative mechanism for burst generation

The positioning of PPN glutamatergic synapses on proximal dendrites near the AIS should maximize their ability to control spike generation. Indeed, at low optical stimulation intensities, SNc DA neurons faithfully followed the pattern of PPN stimulation, even at high frequencies. This behavior

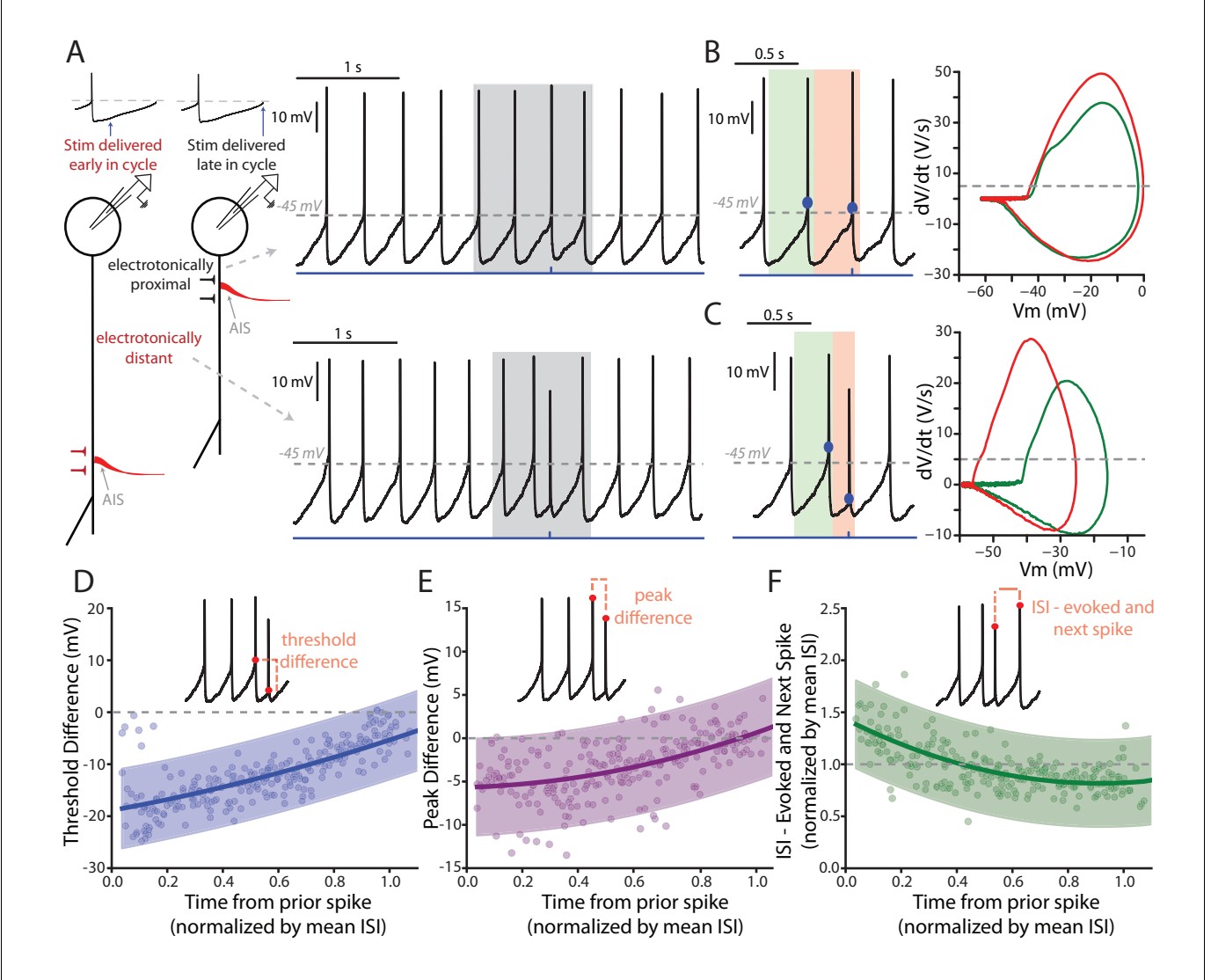

**Figure 5.** Features of PPN-evoked spikes depend on the phase of SNc DA neuron pacemaking cycle. (**A**) Schematic representation of two points in the pacemaking cycle. As the cell nears threshold towards the end of pacemaking cycle it 'tightens up' in preparation of spiking. This manifests as an apparent shortening of the distance between the source of AP generation, the AIS, and the somatic recording electrode due to a longer length constant. Consequently, the evoked spike (Top) appears qualitatively similar to the spontaneously generated APs. In contrast, early in the oscillation cycle the cell is particularly 'leaky', resulting in a small length constant and consequently a spike that appears qualitatively different when recorded at the soma (Bottom). (**B**) Comparison of the PPN evoked spike (red) to the spike preceding the evoked spike (green). Representative spikes are expanded from the shaded region in A. Phase plots (Right), generated from respective green and red shaded regions (left), for the two spikes appear similar, with nearly identical thresholds (determined at the point when the dV/dt exceeds 5 V/S – that is the gray dashed line in the phase plot). (**C**) Similar to (**B**), with the primary difference being the point in the pacemaking cycle where the stimulus occurred. There is a significant shift in the threshold of the evoked spike, as observed by a leftward shift in its associated phase plot (Right, red). (**D–F**) Summaries of three properties of the evoked spike: threshold difference (in comparison to the preceding spike), peak difference (in comparison to the preceding spike), and inter-spike interval between evoked spike and the next spontaneously generated AP. Oscillation phase is represented as time to the stimulus from the preceding spike, normalized by the mean ISI for all the spikes preceding the stimulus (averaging window = 4 s). In (**D**) and (**E**) during the early phase of the oscillation (near t = 0) there is a large deviation in measured spike threshold and spike peak in comparison to the preceding spike. This difference diminishes at t approach 1. In (**F**), except for at the earliest time points the normalized ISI values largely cluster around 1, indicating a resetting of the pacemaking cycle. Fit lines are second-order polynomials, with shaded areas representing 95% confidence intervals.

DOI: https://doi.org/10.7554/eLife.30352.017

The following source data and figure supplements are available for figure 5:

**Source data 1.** Raw data points for panels D, E, and F.

DOI: https://doi.org/10.7554/eLife.30352.020

*Figure 5 continued on next page*

*Figure 5 continued*

**Figure supplement 1.** Effect of SK inhibition on PPN stimulation of SNc DA neurons.

DOI: https://doi.org/10.7554/eLife.30352.018

**Figure supplement 1—source data 1.** Raw data points for panels D and E.

DOI: https://doi.org/10.7554/eLife.30352.019

suggests that PPN synapses near the AIS are capable of driving spike generation independently of NMDARs. The AIS origin of the PPN evoked spikes was consistent with several features of the somatic recordings, including the dependence of the nominal spike threshold and amplitude on the phase of the pacemaking cycle and the faithful propagation of evoked spikes down the axon. Although focal optical stimulation of dendrites that did not bear the AIS might have evoked dendritic spikes that would have manifested greater dependence on the pacemaking cycle, it remains to be determined whether individual PPN axons have terminal fields that are restricted to a single dendrite or diverge to contact several dendrites, including the AIS bearing one.

The degree of PPN convergence on individual SNc DA neurons also is uncertain. Optogenetic stimulation artificially synchronizes spiking in PPN axons, producing a temporal summation of synaptic inputs that may not occur in vivo. This could lead to an over-estimation of the ability of PPN to control the patterning of SNc DA neuron spiking, particularly bursting. However, it is possible that individual SNc DA neurons are innervated by a small number of PPN axons; in this case, synchronization of inputs becomes much less of a concern. The steep relationship between optical stimulus intensity and SNc DA neuron spike probability is consistent with this possibility. This kind of mapping between PPN and SNc would provide an explanation for the variability in SNc burst patterns observed in vivo (*Grace and Bunney, 1984b*; *Hyland et al., 2002*). However, to definitively answer this question, anatomical approaches will be needed. One possibility is single cell mapping experiments using tracer fills of individual PPN axons. Alternatively, a rabies-based retrograde tracing could be employed by generating a sparse starter populations within the SNc (*Wickersham et al., 2013*).

Regardless, these experiments demonstrate that PPN glutamatergic synapses are capable of determining the precise timing of SNc DA neuron spikes that are propagated down the axon to target structures, like the striatum. Moreover, PPN synapses were capable of driving precisely structured bursts, like those recorded in vivo (*Grace and Bunney, 1984b*). This result is consistent with previous studies showing that focal application of glutamate to proximal dendrites, as well as local electrical stimulation of glutamatergic axons, was capable of producing spike bursts (*Blythe et al., 2007*, *2009*).

Thus, there appear to be two general mechanisms by which spike bursts can be generated in SNc DA neurons. In contrast to the PPN-driven mechanism, previous studies have shown that local application of glutamate can evoke bursts that depend upon activation of NMDARs (*Johnson and Wu, 2004*; *Deister et al., 2009*). In vivo, SNc DA neuron burst spiking can be attenuated by NMDAR antagonists (*Charlety et al., 1991*; *Overton and Clark, 1992*; *Smith and Grace, 1992*; *Chergui et al., 1993*) or genetic deletion of NMDARs (*Zweifel et al., 2009*). Similarly, stimulation of the STN in vivo increases burst firing in SNc DA in an NMDA-dependent (*Smith and Grace, 1992*; *Chergui et al., 1994*). The bursts generated in this way harness the intrinsic oscillatory mechanisms of dendrites (*Deister et al., 2009*; *Kuznetsova et al., 2010*; *Ha and Kuznetsov, 2013*). The dependence upon intrinsic oscillatory mechanisms will undoubtedly allow pacemaking and ongoing synaptic input, particularly GABAergic synaptic activity, to influence the timing of bursts, contrasting it with the 'short-circuiting' of these processes by PPN synapses. Another interesting feature of the NMDAR-dependent burst is its dependence upon the voltage-dependence of $Mg^{2+}$ block. Using the dynamic clamp technique, *Deister et al. (2009)* showed that removal of NMDAR voltage sensitivity associated with $Mg^{2+}$ block abolished burst firing. Given that tri-heteromeric GluN1-GluN2B-GluN2D NMDARs found at PPN synapses have lower $Mg^{2+}$ sensitivity (*Huang and Gibb, 2014*), it could be that more distally located NMDARs have a different subunit composition that enhance their ability to promote bursts. Lastly, it is unclear to what extent extra-synaptic NMDARs play a role in burst generation. Given the differences in postsynaptic signaling by synaptic and extrasynaptic NMDARs (*Hardingham and Bading, 2010*; *Paoletti et al., 2013*), the differential engagement by the two different modes of burst generation could have important long-term consequences.

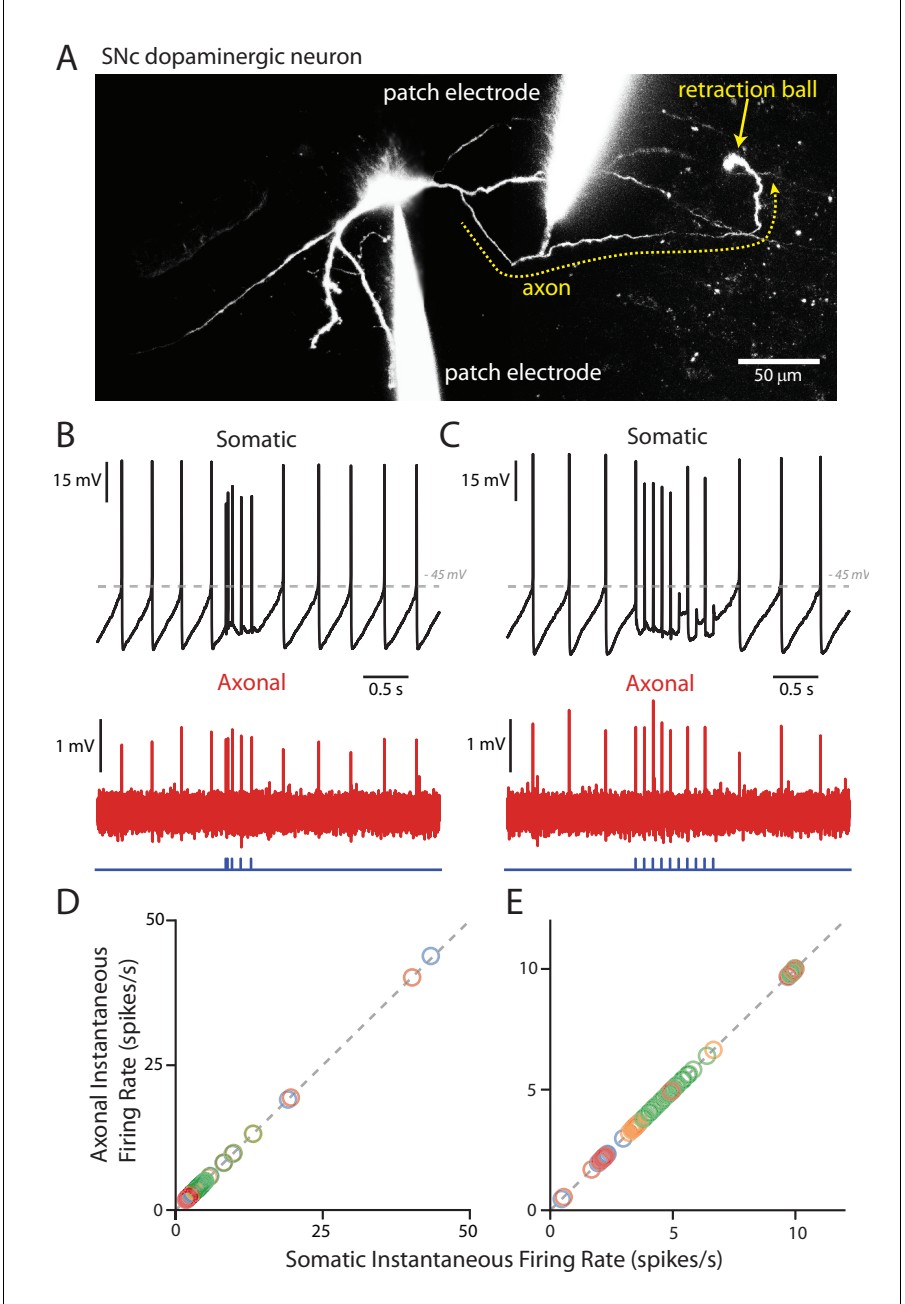

**Figure 6.** PPN-evoked spikes reliably invade the axon. (**A**) Reconstruction of an example cell where two recording electrodes are clearly visible. Paired recording were made in order to simultaneously measure evoked spikes at the soma and at the axon. Axons (yellow dashed line) were identified based upon the presence of a retraction ball (yellow arrow) following the filling of the cell with dye via the somatic electrode. After identification, loose-seal recordings were made of spike activity in the axon. (**B**) Example recording of somatic whole-cell (Top) and axonal loose-seal (Bottom) recordings of spontaneous and stimulated (5, 1 ms stimuli with variable inter-stimulus interval protocol) action potentials. Spikes recorded at the soma are mirrored by events in the axon. (**C**) Similar to (**B**), except for the application of a different stimulation protocol (10, 1 ms stimuli with an inter-stimulus interval of 100 ms). (**D–E**) Plot of the instantaneous spike rate (spikes/s) of APs recorded in the axon as a function of the instantaneous spike rate (spikes/s) of APs recorded in the soma. Points fall along a linear, 1:1 relationship (gray dashed lines), indicating reliable representation in the axonal recordings of spike events also recorded in the soma. (**D**) are data points from the variable inter-stimulus interval protocol; (**E**) are data points from the 10 Hz stimulation protocol. Different colors represent different cells (n = 4).

DOI: https://doi.org/10.7554/eLife.30352.021

*Figure 6 continued on next page*

*Figure 6 continued*

The following source data is available for figure 6:

**Source data 1.** Raw data points for panels D and E.

DOI: https://doi.org/10.7554/eLife.30352.022

## Functional consequences of PPN control of SNc

There are several lines of evidence suggesting that the PPN exerts a strong control over DA neuron spiking in vivo. For example, stimulation of the PPN in vivo produces a significant increase in bursting in ventral tegmental area (VTA) DA neurons and dopamine release in the nucleus accumbens (*Floresco et al., 2003*). Moreover, lesioning the PPN disrupts dopamine-dependent learning in a variety of behavioral tasks (*Inglis et al., 2000*; *Alderson et al., 2008*; *Bortolanza et al., 2010*; *Jimenez-Martin et al., 2015*; *Syed et al., 2016*). Recordings in behaving animals have shown that PPN neurons code for predicted reward value, reward magnitude and stimulus salience (*Pan and Hyland, 2005*; *Kobayashi and Okada, 2007*; *Okada et al., 2009*; *Norton et al., 2011*; *Okada and Kobayashi, 2013*). Often, the activity in PPN precedes that in SNc (*Pan and Hyland, 2005*; *Kobayashi and Okada, 2007*; *Okada et al., 2009*), raising the possibility that signals from the PPN are critical for the computations performed by DA neurons.

In addition to reward signaling, recent work has implicated PPN glutamatergic neurons in the control of movement gated by the striatum (*Roseberry et al., 2016*). Our results show that this activity provides a potent excitation of SNc DA neurons that may be important to modulating striatal circuits controlling movement sequences. This inference is consistent with the observation that phasic activity in SNc DA neurons is temporally correlated not only with the outcomes of actions, but action itself (*Howe and Dombeck, 2016*). Precisely what is being coded by PPN activity influencing SNc and how this translates into the pattern of SNc spiking remains to be determined by in vivo experiments (*Hong and Hikosaka, 2014*). Nevertheless, it is tempting to speculate that the basal ganglia control of PPN glutamatergic neurons described by *Roseberry et al. (2016)* is fed back to the SNc and broadcast to the striatum, allowing complex movement sequences to be executed – a capacity that is lost in PD patients with degeneration of SNc DA neurons (*Hernández et al., 2015*).

## Summary

*O*ur studies identify a novel mechanism for burst generation in SNc DA neurons. Glutamatergic innervation of the SNc by the PPN was found to preferentially target proximal regions of the SNc dendritic tree, near where the AIS of these cells generally originates, placing these synapses in a favorable location to drive *spiking*. In contrast to the previously described mechanism involving the cooperation of intrinsic oscillatory activity with NMDAR activation to generate bursts in SNc neurons, activation of PPN fibers was capable of directly patterning SNc neuron spiking independently of regular pacemaking activity and without the need to engage NMDARs. Further examination is required to determine whether this mechanism generalizes to other sources of glutamatergic input to SNc neurons. Nevertheless, this work indicates the PPN is a likely source of extrinsic control of SNc neuron firing *during* goal-directed behavior.

# Materials and methods

## Animals

Male and female C57Bl/6J (Jackson Laboratory, Bar Harbor, ME, Stock #000664) or DAT-Cre/Ai14-tdTomato (on a C57Bl/6J background) mice were used. The latter were generated in-house using DAT-Cre (*Slc6a3$^{tm1.1(cre)Bkmn}$* – Jackson Laboratory, Stock #006660) and Ai14 (*Gt(ROSA)26Sor$^{tm14}$ $^{(CAG-tdTomato)Hze}$* – Jackson Laboratory, Stock #007908) mice. All experiments were performed in accordance with protocols reviewed and approved by the Northwestern Institutional Animal Care and Use Committee and NIH guidelines.

## Stereotaxic injections

Stereotaxic injections were performed when animals were between P16 and P25 days old. Animals were anesthetized with an isoflurane precision vaporizer (Smiths Medical PM, Inc., Norwell, MA) and placed in a stereotaxic frame (David Kopf Instruments, Tujunga, CA). The distance between bregma and lambda was measured and used to adjust the following stereotaxic coordinates: PPN – AP: −4.4, ML: 1.25, DV: 3.5; STN – AP: −1.8, ML: 1.4, DV: 4.5. A small hole was drilled using a micro drill bit (Fine Science Tools, Foster City, CA) and a calibrated glass pipette pulled on a P-97 Sutter Instruments (Novato, CA) puller was used to inject 40–60 nL of either AAV9.CAG.hChR2, AAV9.hSyn. hChR2, or AAV9.Syn.Chronos (Addgene 20938M, Addgene 26973P, or Addgene 62726, respectively, supplied by University of Pennsylvania Vector Core) at one of these locations. Animals were sacrificed 10–20 days post injection.

## Slice preparation

Mice were anaesthetized with a mixture of ketamine (50 mg/kg) and xylazine (4.5 mg/kg) and intracardially perfused with ice-cold high-sucrose, high-magnesium artificial cerebrospinal fluid (aCSF) containing (in mM): 50 NaCl, 2.5 KCl, 25 $NaHCO_3$, 1.25 $NaH_2PO_4$, 1 $CaCl_2$, 10 $MgCl_2$, 25 glucose, pH 7.3 (~310 mOsm/L). The brain was then removed, sectioned in to 220–275 μm coronal or parasagittal slices using a Leica VT1200 S vibratome (Wetzlar, Germany), and allowed to recover at room temperature for at least 30 min in aCSF containing (in mM): 82.5 NaCl, 2.5 KCl, 25 $NaHCO_3$, 1.25 $NaH_2PO_4$, 1.5 $CaCl_2$, 5.5 $MgCl_2$, 25 glucose, pH 7.3 (~310 mOsm/L). All solutions were oxygenated with a mixture of 95% O2/5% CO2.

## Electrophysiology

Slices were transferred to a recording chamber continuously perfused with warm (33–35 C), oxygenated aCSF containing (in mM): 125 NaCl, 2.5 KCl, 25 $NaHCO_3$, 1.25 $NaH_2PO_4$, 2 $CaCl_2$, 1 $MgCl_2$, 25 glucose, pH 7.3 (~310 mOsm/L). Cells were visualized on an Olympus BX51 upright microscope equipped with an Olympus LUMPFL 60 × 1.0 NA water-dipping objective lens using a Thorlabs 1545M CMOS USB camera and Micro-Manager open source microscopy software (*Edelstein et al., 2001*). Stage movement, objective lens focus, and manipulator XYZ movement was controlled, respectively, by: FM-380 shifting stage, Olympus axial focus module, and manipulators (Luigs and Nuemann GmbH; Ratingen, Germany)

Patch pipettes were pulled from thick-walled borosilicate glass on a Sutter P-1000 puller. Pipette resistance was typically 3–4 MΩ, except for whole cell pacemaking and cell-attached axon recordings where pipette resistance typically was 8–15 MΩ. Several different internal solutions were used, depending on the experiment being performed. For whole-cell voltage-clamp experiments pipettes were filled with a cesium-based internal containing (in mM): 120 $CsMeSO_3$, 15 CsCl, 10 HEPES, 0.2 EGTA, 3 ATP-Mg, 0.3 GTP-Na, 10 TEA-Cl, 1.9 QX314-Cl. For whole-cell voltage-clamp calcium imaging experiments the same internal, absent EGTA, was supplemented with 100 μM Fluo-4 or Fura-2 and 25 μM Alexa 568. For whole-cell current-clamp experiments pipettes were filled with a potassium-based internal containing (in mM): 135 $KMeSO_4$, 5 KCl, 5 HEPES, 0.05 EGTA, 10 phosphocreatine-di(tris), 2 ATP-Mg, 0.5 GTP-Na as well as 25 μM Alexa 568. Perforated-patch experiments were performed with pipettes front-filled with a solution containing (in mM): 126 $KMeSO_4$, 14 KCl, 10 HEPES, 1 EGTA, 0.5 $CaCl_2$, $3MgCl_2$ and then back filled with the same solution containing 20 μg/mL gramicidin-D. Loose-seal cell-attached recordings were made with pipettes filled with 145 mM NaCl, 10 mM HEPES, and 25 μM Alexa 568 (ThermoFischer Scientific, Waltham, MA). All internals had a pH of 7.25–7.3 (with either 1 M CsOH, KOH, or NaOH) and an osmolarity of 280–300 mOsm/L.

Electrophysiological recordings were obtained using a Multiclamp 700B amplifier. Signals were filtered at 4–20 kHz and digitized at 10–50 kHz. For voltage clamp experiments access resistance was monitored throughout the experiment. Cells in which access deviated from baseline by more than 20% were discarded. For perforate patch experiments, cells were left to perforate until the spike height reached roughly 0 mV before data collection began. Rapid jumps in the observed voltage to positive (>0 mV) values were used as exclusion criterion due to break-in. The liquid junction potential for the cesium, potassium, and perforated internals in recording aCSF were 5.9 mV, 7 mV, and 5.1 mV respectively, and were corrected for during data analysis.

## Pharmacology

A number of different pharmacological agents were used. Unless otherwise noted, drugs were purchased from R and D Systems (Minneapolis, MN) or Abcam (Cambridge, MA) and were prepared according to manufacturer instructions. They are listed here, along with their working concentration: NBQX (5 µM), (R)-CPP (5 µM), SR95531 (10 µM), mecamylamine hydrochloride (10 µM), tetrodotoxin (1 µM), 4-aminopyridine (100 µM), apamin (200 nM), ifenprodil (5 µM), TCN 201 (10 µM), A 841720 (0.1 µM), MTEP hydrochloride (0.5 µM), philanthotoxin 74 (5 µM), glycine (250 µM), D-serine (250 µM), CIQ (10 µM; provided the Traynelis lab and Brandt Labs, Atlanta, GA).

## 2PLSM imaging and photostimulation

Two-photon laser-scanning microscopy (2PLSM) was performed using an Ultima LSM system (Prairie Technologies, Middleton, WI). The 2P excitation source was a Chameleon-Ultra series tunable (690–1040 nm) Ti:sapphire laser system (Coherent Laser Group, Santa Clara, CA). Alexa and Fluo-4 dyes were excited using 820 nm (80 MHz pulse repetition frequency and ~140 fs pulse duration) excitation, while Fura-2 was imaged at 780 nm. Laser power attenuation was achieved with two Pockels' cell electro-optic modulators (models M350-80-02-BK and M350-50-02-BK, Con Optics, Danbury, CT) controlled by PrairieView 5.0–5.3 software. The two cells were aligned in series to provide enhanced modulation range for fine control of the excitation dose (0.1% steps over five decades), to limit maximum power, and to serve as a rapid shutter during line scan acquisitions. The fluorescence emission was collected by non–de-scanned photomultiplier tubes (PMTs). Green channel (490–560 nm) signals were detected by a Hamamatsu H7422P-40 select GaAsP PMT, which was protected during blue laser exposures with a Uniblitz DSS10B-1-T-0 shutter (Vincent Associates). Red channel (580–630 nm) signals were detected by a Hamamatsu R3982 side on PMT. Cell visualization during laser scanning was made possible by a Dodt-tube-based transmission detector with a mirror routing the laser to a Hamamatsu R3982 side on PMT (Prairie Technologies; Middleton, WI). Scanning signals were sent and received by the PCI-NI6110 analog-to-digital converter card in the system computer (National Instruments, Austin, TX). Scanned images were built up pixel by pixel (dwell time: 10–12 µs), with PMT anode current to voltage conversion and sampling fixed in 0.4 µs increments. For calcium imaging, line scans were performed along 5–10 µm sections of dendrite (6 ms and 512 pixels per line). Cells were allowed to fill for a minimum of 30 min to allow for dye equilibration. Calcium fluorescence signals were background-subtracted and normalized by a baseline fluorescence ($f_0$). Calcium transients were quantified as the area under the transient.

One-photon (1P) photostimulation was performed using either a Prairie Aurora Launch (473 nm, 50 mW rated laser with AOTF intensity control) or a Prairie Helios laser launch (Coherent OBIS 473 nm laser). The launch was coupled to the Ultima scan head via a metal clad fiber optic cable. The launch optics were designed to provide ~1 um spot stimulation at the focal plan of the 60x/1.0NA objective lens; additional optics allowed for this spot size to be increased to ~10 um. Generally, a 1.0ND filter was employed to reduce the maximal power at the sample from ~18 mW to ~1.8 mW. Laser power was further controlled by the PrairieView software. A second pair of galvanometer mirrors within the Ultima scan head allowed for multiple stimulation points within the focal plane of interest. Full-field photostimulation was provided by either a pE-100 470 nm LED (CoolLED via Tek5 Systems, Yorktown Heights, NY) or an Excelitas Excite LED110 four-LED, coupled to scanning system via a Lumatec 3 mm liquid light guide via the Olympus BX-51 WIF rear epi-fluorescence port. For blue-light stimulation, a Chroma 39002 ET eGFP large (BX2) Olympus filter cube was used. The LED was remotely synchronized and activated by a TTL signal from the PrairieView software. The maximum exposure field of view with the 60x/1.0NA objective lens was ~415 µm.

For sCRACM experiments, the point spread function of the blue laser was estimated by moving the nominal site of stimulation away from the dendrite (*Figure 3—figure supplement 1*). Laser power was calibrated based on this same procedure. For RuBi-glutamate uncaging experiments, RuBi-glutamate (2 mM) was superfused (0.4 ml/hr) using a system of syringe pumps (World Precision Instruments, Sarasota, FL) and a multi-barreled perfusion manifold fitted with a small-volume mixing tip that allowed rapid switching between solutions (Cell MicroControls, Norfolk, VA). In all photostimulation experiments, light pulses were limited to 1 ms, with the power calibrated based on achieving physiological responses within the scope of the respective experiment.

## Fixed tissue preparation and imaging

Fixed tissue was prepared by first perfusing anaesthetized animals with phosphate buffered saline (PBS, Sigma-Aldrich, St. Louis, MO) followed by 4% paraformaldehyde (PFA, diluted with PBS from 16% stock, Electron Microscopy Sciences, Hatfield, PA). The brain was then removed and allowed to further fix in PFA overnight, and then washed and stored in PBS. Brains were sectioned in to 100 µm parasagittal slices on a Leica VT1200 S vibratome. Sections were mounted on with ProLong Diamond (ThermoFischer Scientific, Waltham, MA) and imaged with an Olympus FV10i confocal laser scanning microscope.

## Data analysis and statistics

Both electrophysiological and imaging data were analyzed using either GraphPad Prism (version 5.0, GraphPad Software), Fiji (*Schindelin et al., 2012*), or custom written python analysis scripts using a number of numerical python packages: pandas (*McKinney, 2010*), SciPy (*Jones et al., 2001*), and statsmodels (*Seabold and Perktold, 2010*). Figures were created with matplotlib (*Hunter, 2007*) or GraphPad Prism and Adobe Illustrator. The code is available at https://github.com/surmeierlab/neurphys (*Galtieri and Estep, 2014*); a copy is archived at https://github.com/elifesciences-publications/neurphys. Data were summarized using box plots showing median values, first and third quartiles, and whiskers at 10th and 90th percentiles. Summary statistics are presented as mean ± standard deviation. Sample $n$ represents the number of neurons collected from brain slices from a minimum of three animals. Sample size was based on prior studies published from our lab and others using similar techniques (*Blythe et al., 2009*; *Sanchez-Padilla et al., 2014*). Statistical analysis was performed with either SciPy, statsmodels, or GraphPad Prism using non-parametric tests (Mann-Whitney U-test of significance or Wilcoxon signed rank test for between or within-subject design experiments, respectively) except where otherwise noted. To correct for multiple comparisons the Holm-Bonferroni method was used, with the reported p-value representing the adjusted p-value. Probability threshold for statistical significance was $p < 0.05$.

## Acknowledgements

This work was supported by grants from the JPB Foundation, the IDP Foundation and NIH (NS 047085). We thank Dr. Harini Lakshminarasimhan for her technical assistance with the fixed-tissue imaging and Sasha Ulrich, Danielle Schowalter, and Marisha Alicea for their management of the various mouse colonies used here.

## Additional information

### Funding

| Funder | Author |
|--------|--------|
| JPB Foundation | Daniel J Galtieri<br>Chad M Estep<br>David L Wokosin<br>D James Surmeier |
| IDP Foundation | Daniel J Galtieri<br>Chad M Estep<br>David L Wokosin<br>D James Surmeier |
| National Institute of Neurological Disorders and Stroke | Daniel J Galtieri<br>Chad M Estep<br>David L Wokosin<br>D James Surmeier |

The funders had no role in study design, data collection and interpretation, or the decision to submit the work for publication.

## Author contributions

Daniel J Galtieri, Conceptualization, Data curation, Software, Formal analysis, Validation, Investigation, Visualization, Methodology, Writing—original draft, Project administration, Writing—review and editing; Chad M Estep, Data curation, Software, Validation, Investigation, Methodology, Writing—review and editing; David L Wokosin, Stephen Traynelis, Resources, Methodology, Writing—review and editing; D James Surmeier, Conceptualization, Resources, Supervision, Funding acquisition, Methodology, Writing—original draft, Project administration, Writing—review and editing

## Author ORCIDs

Stephen Traynelis  http://orcid.org/0000-0002-3750-9615
D James Surmeier  http://orcid.org/0000-0002-6376-5225

## Ethics

Animal experimentation: All experiments were performed in strict accordance with the guidelines set by the Guide for the Care and Use of Laboratory Animals of the National Institutes of Health. All animals were handled according to approved Institutional Animal Care and Use Committee protocols (IS00001185) of Northwestern University. All procedures were performed under isoflurane or ketamine/xylazine anesthesia, and every effort was made to minimize suffering.

## Decision letter and Author response

Decision letter https://doi.org/10.7554/eLife.30352.024
Author response https://doi.org/10.7554/eLife.30352.025

# Additional files

## Supplementary files

• Transparent reporting form
DOI: https://doi.org/10.7554/eLife.30352.023

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
