## [Decision Letter]

Thank you for submitting your article "Pedunculopontine glutamatergic neurons control spike patterning in substantia nigra dopaminergic neurons" for consideration by *eLife*. Your article has been reviewed by three peer reviewers, and the evaluation has been overseen by a Reviewing Editor and Gary Westbrook as the Senior Editor. The following individuals involved in review of your submission have agreed to reveal their identity: John T Williams (Reviewer #2); Charles Wilson (Reviewer #3). The reviewers have discussed the reviews with one another and the Reviewing Editor has drafted this decision to help you prepare a revised submission.

Summary:

The reviewers and the BRE find that this study describes an important input/ mechanism that could mediate phasic responses of DA neurons in vivo. However, there is no direct testing that the mechanism presented mediates bursting in vivo. Everyone agreed that in vivo experiments, although important, would take too long for this study. So the consensus was not to ask for new experiments, but instead for a change in language so that the paper does not claim that this is the mechanism for DA neuron bursting in vivo. The authors could still suggest that this may mediate phasic responses of DA neurons in the discussion, and also suggest that this is a novel mechanism which may mediate responses to stimuli or movement. This summary should guide your response to the individual reviewer comments.

Below are the detailed comments of the reviewers that may help prepare the revised manuscript.

Reviewer #1:

In this manuscript, the authors propose that SN DA neuron burst firing is regulated by PPN input, with the SNDA neurons following optogenetic activation of the PPN. The authors further propose that burst firing may be driven in a spike-to-spike manner by patterned burst firing of the PPN. While this is an intriguing finding, the suggestion that this underlies burst firing pattern of SNDA neurons is far from validated. First, with optogenetic stimulation, all of the PPN-SN fibers are activated simultaneously; a situation that does not exist in vivo. Secondly, it cannot account for the ability of NMDA blockade in vivo in preventing burst firing of SNDA neurons. Third, burst firing has never been shown to occur simultaneously across neurons in the SN (which one would expect from a burst PPN input), but occurs with a range of ISIs, burst length, etc. And finally, and importantly, the timing of the burst (length, ISI) of PPN neurons does not follow the characteristics of burst firing in DA neurons, with substantially longer ISIs. Therefore, while this is clearly a powerful and important input that is well-characterized here both anatomically and electrophysiologically, to suggest that it underlies burst firing would require substantially more data, including tonic activation of the PPN (e.g., with DREADDs or pharmacologically), demonstration in vivo that bursting in PPN parallels temporally burst firing in SNC, and the ability to prevent burst firing in vivo (including NMDA-driven burst firing) to be able to support such a specific mechanism. I would suggest the authors focus on the powerful PPN-SNC connection, the extensive characterization of the glutamatergic mechanisms, and its implications for driving normal firing or modulating output, rather than trying to force their data into a tenuous explanation of burst firing that is strongly countered by a rather substantial body of data.

Subsection “PPN glutamatergic synapses were capable of spike patterning” – There is an explanation for the truncated action potentials that has been published before. In fact the explanation provided in the next paragraph is identical to that demonstrated 30 years ago. Specifically, it has been shown (see early Grace papers) that antidromic activation of DA neurons often triggers only an IS spike and not a full IS-SD spike. This was shown using in vivo intracellular recording that with somatic depolarization, a full spike is triggered. The explanation is that during normal firing, the soma is first depolarized by the pacemaker conductance, allowing the IS spike to spread into the dendrites to trigger dendritic spiking. However, with AD, the soma is hyperpolarized, and the is spike cannot spread to depolarize the dendritic tree.

Reviewer #2:

This manuscript describes the glutamate input from the peduculopontine nucleus (PPN) to dopamine neurons in the substantia nigra compacta in the mouse. Whole cell, perforated patch and loose cell attached recordings were made in combination with optogenetic activation of PPN axons, electrical stimulation and glutamate uncaging to pharmacologically define the subunit composition of the AMPA and NMDA receptors. The results show that GluA2 containing subunits make up AMPA receptors and NMDA receptors contained GluN1, GluN2B and GluN2C/D subunits at this synapse. The most interesting observation is that the inputs from PPN more or less selectively innervate the cell body and proximal dendrites. This pattern of innervation resulted in reliable activation of action potentials in the axon and cell body. The conclusion was that the AMPA receptor activation arising from the PPN alone is capable of driving burst activity in dopamine neurons.

1). This is a concise manuscript that describes comprehensively one key excitatory input onto dopamine neurons in the substantia nigra. As such this is an important contribution.

2) The experiments use appropriate methods and the results support the conclusions.

3) A series of important controls are included that support the most interesting observation that the inputs from the PPN selectively innervate proximal dendrites and therefore are key regulators of soma/axonal excitability. As far as I know, this is the first demonstration of a selective distribution of excitatory afferents on dopamine neurons and is an important observation.

Reviewer #3:

This is an original and unexpected finding on the synaptic origins of burst firing in the dopaminergic neurons of the substantia nigra. Studies in vivo had implicated NMDA receptors as essential, but were not successful at determining the nigral afferent pathway that was directly responsible for bursts. It is important because burst firing in dopaminergic neurons has been hypothesized to be a final common pathway for the internal model of reward prediction. It seems unlikely that the dopaminergic neuron receives direct sensory and memory-related inputs required to synthesize a reward prediction. It is usually assumed that these signals are synthesized elsewhere and combined at the level of the dopaminergic neuron. This paper suggests that the actual combination of sensory and memory information may occur at an even more unexpected location, pedunculopontine nucleus of the pontine reticular formation.

The experiments use optogenetic activation of afferents from the pedunculopontine nucleus to trigger bursts in dopaminergic cells. The authors do a good job of verifying their stimulus and characterizing the contribution of the various glutamate receptor subtypes.

A key part of the finding is the demonstration that the synapses responsible are located proximally. For this they use focal optogenetic stimulation. They argue that the proximal location of the synaptic input is key for the generation of burst firing, because fast proximal synchronous activation of the synapses could drive spiking at high frequencies. Spikes could be driven with this stimulus at rates and in patterns that could not be followed by the entire somatodendritic membrane, suggesting that the proximity of the input to the axonal spike generator allowed firing at rates the soma could not sustain. They showed that these partial spikes were propagated down the axon. But it was surprising that fast spiking driven by this stimulus did not depend on NMDA receptors.

The experiments are well designed and skillfully executed. The figures and the writing are very clear and illustrative. I think the paper is very convincing.

My only concern is that the authors don't explain how a burst in vivo can actually be generated in their scheme. It seems essential that the neurons in the pedunculopontine nucleus fire in synchrony at high frequency, in a style similar to that used in the experiment, to drive a burst in the dopaminergic neurons. The dopaminergic neurons under these circumstances should also fire synchronously. Is this what happens? Or could single PPN neurons fire bursts which are sufficient to trigger identical bursts, spike-for-spike in some small subset of dopaminergic neurons. In that case asynchronous bursts in PPN neurons could produce asynchronous bursting in the SNC. But then single PPN neurons must be effective at triggering spikes. Is there any indication of how powerful a single PPN neuron is?

One implication of this that the authors do not discuss is the function of the dopaminergic neuron in burst formation. This is, of course, the reason for studying all this in the first place. Their scheme seems to make the dopaminergic cell a mere relay for reward-prediction bursts generated in the PPN. This of course does not explain how the brain can do reward prediction, it merely passes the buck to the PPN. How closer does this get us to understanding reward prediction? Can the PPN synthesize reward prediction error?

---

## [Author Response]

Summary:The reviewers and the BRE find that this study describes an important input/ mechanism that could mediate phasic responses of DA neurons in vivo. However, there is no direct testing that the mechanism presented mediates bursting in vivo. Everyone agreed that in vivo experiments, although important, would take too long for this study. So the consensus was not to ask for new experiments, but instead for a change in language so that the paper does not claim that this is the mechanism for DA neuron bursting in vivo. The authors could still suggest that this may mediate phasic responses of DA neurons in the discussion, and also suggest that this is a novel mechanism which may mediate responses to stimuli or movement. This summary should guide your response to the individual reviewer comments.Below are the detailed comments of the reviewers that may help prepare the revised manuscript.

It was not our intention to give the impression that we were describing THE mechanism responsible for burst spiking in vivo. The current view of burst spiking in SNc DA neurons is that NMDARs are necessary. Our point was simply to provide an example where this wasn’t the case. In the revision, we have attempted to make this point clear and to underscore the limits of our experimental approach.

Reviewer #1:In this manuscript, the authors propose that SN DA neuron burst firing is regulated by PPN input, with the SNDA neurons following optogenetic activation of the PPN. The authors further propose that burst firing may be driven in a spike-to-spike manner by patterned burst firing of the PPN. While this is an intriguing finding, the suggestion that this underlies burst firing pattern of SNDA neurons is far from validated.First, with optogenetic stimulation, all of the PPN-SN fibers are activated simultaneously; a situation that does not exist in vivo.

Optogenetic stimulation induces synchrony in a subset of PPN axons; it does not activate ALL PPN axons. Nevertheless, it is true that it does induce a degree of synchrony that is unlikely to occur in vivo. But, this is also true other methods that have been used to probe the mechanisms controlling spike generation in SNc neurons (e.g., iontophoresis of glutamate or electrical stimulation of afferent fibers). As shown in Figure 4—figure supplement 1, we attempted to alleviate this concern by grading the optical stimulation intensity to estimate the extent of the PPN fiber recruitment that was necessary to reliably evoke spikes. As seen in this figure, this relationship was very steep and plateaued at low photon doses, suggesting that only a relatively small number of fibers was necessary for spike generation. The revision has been edited to make this point more clearly. More importantly, the paper makes the point that PPN synapses have the *potential* to control the spiking of SNc neurons by virtue of their preferential, proximal dendritic location, near the axon initial segment; this positioning allows them to produce burst spiking without NMDARs.

Secondly, it cannot account for the ability of NMDA blockade in vivo in preventing burst firing of SNDA neurons.

The available evidence suggest that NMDAR blockade or genetic removal reduces, but does not eliminate burst spiking in SNc DA neurons of mice in an open field. There is no compelling evidence that NMDAR blockade affects burst spiking generated by salient or rewarding stimuli. In fact, the work of Zweifel et al,. (2009) argues that many behaviors that recent work has shown to be correlated with burst spiking in SNc DA neurons – like the initiation of movement (Howe and Dombeck, 2016) – are unaffected by genetic deletion of NMDARs. Our view, which has been significantly expanded in the revision, is that there are several ways in which bursts are generated in SNc DA neurons. What our work does it to broaden this ‘burst canvas’ to include NMDAR-independent mechanisms. Again, this does not exclude NMDAR-dependent mechanisms.

Third, burst firing has never been shown to occur simultaneously across neurons in the SN (which one would expect from a burst PPN input), but occurs with a range of ISIs, burst length, etc.

There is no reason to think that all of the glutamatergic PPN neurons spike in synchrony. If only a few PPN neurons control each SNc DA neuron, then there can be a high degree of heterogeneity in the burst spiking of SNc DA neurons. This point is made in the revision.

And finally, and importantly, the timing of the burst (length, ISI) of PPN neurons does not follow the characteristics of burst firing in DA neurons, with substantially longer ISIs.

While the observation made by Pan and Hyland (2005) indicate that PPN neurons fire at much higher frequencies in response to sensory cues than do SNc neurons, work by others has shown a much closer overlap in the spiking patterns of PPN and SNc neurons encoding predicted rewards and reward delivery (Hyland et al., 2002; Kobayashi and Okada, 2007; Okada et al., 2009; Hong and Hikosaka, 2014). That said, given the uncertainty about the extent of PPN axon convergence on SNc DA neurons, there need not be any relationship between burst patterning in PPN and SNc – other than onset. Again, we have attempted to make this point in the revision.

Therefore, while this is clearly a powerful and important input that is well-characterized here both anatomically and electrophysiologically, to suggest that it underlies burst firing would require substantially more data, including tonic activation of the PPN (e.g., with DREADDs or pharmacologically), demonstration in vivo that bursting in PPN parallels temporally burst firing in SNC, and the ability to prevent burst firing in vivo (including NMDA-driven burst firing) to be able to support such a specific mechanism. I would suggest the authors focus on the powerful PPN-SNC connection, the extensive characterization of the glutamatergic mechanisms, and its implications for driving normal firing or modulating output, rather than trying to force their data into a tenuous explanation of burst firing that is strongly countered by a rather substantial body of data.

SNc DA neurons receive glutamatergic input from several sources other than PPN: cortical pyramidal neurons, subthalamic nucleus neurons, superior colliculus neurons to name a few. We make no claims about the ability of these inputs to generate NMDAR-dependent bursts. Our only point is that PPN synapses, by virtue of their unusual positioning close to the axon initial segment, are capable of patterning spiking in SNc DA neurons without the engagement of NMDARs. Previous work (Grace and Bunney, 1983; Häusser et al., 1995; Blythe et al., 2009) is consistent with this proposition.

Subsection “PPN glutamatergic synapses were capable of spike patterning” – There is an explanation for the truncated action potentials that has been published before. In fact the explanation provided in the next paragraph is identical to that demonstrated 30 years ago. Specifically, it has been shown (see early Grace papers) that antidromic activation of DA neurons often triggers only an IS spike and not a full IS-SD spike. This was shown using in vivo intracellular recording that with somatic depolarization, a full spike is triggered. The explanation is that during normal firing, the soma is first depolarized by the pacemaker conductance, allowing the IS spike to spread into the dendrites to trigger dendritic spiking. However, with AD, the soma is hyperpolarized, and the is spike cannot spread to depolarize the dendritic tree.

We appreciate the reviewer pointing out this reference. It has been added to the revision. However, it is important to point out that what was shown in our paper were synaptically generated, orthodromic spikes, not antidromically evoked spikes. Moreover, the IS-SD spike differences with antidromically evoked spikes was shown much earlier in other neurons by Eccles and colleagues.

Reviewer #2:This manuscript describes the glutamate input from the peduculopontine nucleus (PPN) to dopamine neurons in the substantia nigra compacta in the mouse. Whole cell, perforated patch and loose cell attached recordings were made in combination with optogenetic activation of PPN axons, electrical stimulation and glutamate uncaging to pharmacologically define the subunit composition of the AMPA and NMDA receptors. The results show that GluA2 containing subunits make up AMPA receptors and NMDA receptors contained GluN1, GluN2B and GluN2C/D subunits at this synapse. The most interesting observation is that the inputs from PPN more or less selectively innervate the cell body and proximal dendrites. This pattern of innervation resulted in reliable activation of action potentials in the axon and cell body. The conclusion was that the AMPA receptor activation arising from the PPN alone is capable of driving burst activity in dopamine neurons.1). This is a concise manuscript that describes comprehensively one key excitatory input onto dopamine neurons in the substantia nigra. As such this is an important contribution.2) The experiments use appropriate methods and the results support the conclusions.3) A series of important controls are included that support the most interesting observation that the inputs from the PPN selectively innervate proximal dendrites and therefore are key regulators of soma/axonal excitability. As far as I know, this is the first demonstration of a selective distribution of excitatory afferents on dopamine neurons and is an important observation.

We thank the reviewer for their time in evaluating our work and for their supportive comments in response to our presented findings.

Reviewer #3:This is an original and unexpected finding on the synaptic origins of burst firing in the dopaminergic neurons of the substantia nigra. Studies in vivo had implicated NMDA receptors as essential, but were not successful at determining the nigral afferent pathway that was directly responsible for bursts. It is important because burst firing in dopaminergic neurons has been hypothesized to be a final common pathway for the internal model of reward prediction. It seems unlikely that the dopaminergic neuron receives direct sensory and memory-related inputs required to synthesize a reward prediction. It is usually assumed that these signals are synthesized elsewhere and combined at the level of the dopaminergic neuron. This paper suggests that the actual combination of sensory and memory information may occur at an even more unexpected location, pedunculopontine nucleus of the pontine reticular formation.The experiments use optogenetic activation of afferents from the pedunculopontine nucleus to trigger bursts in dopaminergic cells. The authors do a good job of verifying their stimulus and characterizing the contribution of the various glutamate receptor subtypes.A key part of the finding is the demonstration that the synapses responsible are located proximally. For this they use focal optogenetic stimulation. They argue that the proximal location of the synaptic input is key for the generation of burst firing, because fast proximal synchronous activation of the synapses could drive spiking at high frequencies. Spikes could be driven with this stimulus at rates and in patterns that could not be followed by the entire somatodendritic membrane, suggesting that the proximity of the input to the axonal spike generator allowed firing at rates the soma could not sustain. They showed that these partial spikes were propagated down the axon. But it was surprising that fast spiking driven by this stimulus did not depend on NMDA receptors.The experiments are well designed and skillfully executed. The figures and the writing are very clear and illustrative. I think the paper is very convincing.My only concern is that the authors don't explain how a burst in vivo can actually be generated in their scheme. It seems essential that the neurons in the pedunculopontine nucleus fire in synchrony at high frequency, in a style similar to that used in the experiment, to drive a burst in the dopaminergic neurons. The dopaminergic neurons under these circumstances should also fire synchronously. Is this what happens? Or could single PPN neurons fire bursts which are sufficient to trigger identical bursts, spike-for-spike in some small subset of dopaminergic neurons. In that case asynchronous bursts in PPN neurons could produce asynchronous bursting in the SNC. But then single PPN neurons must be effective at triggering spikes. Is there any indication of how powerful a single PPN neuron is?

First of all, we thank the reviewer for their encouragement and positive comments. Second, the reviewer raises an excellent question – one for which we don’t have an answer. We’re grappling with strategies that would allow us to get at an answer but we’re not there yet. What we have done in the revision is raise the broader question in the Discussion section and pose some potential strategies for answering it.

One implication of this that the authors do not discuss is the function of the dopaminergic neuron in burst formation. This is, of course, the reason for studying all this in the first place. Their scheme seems to make the dopaminergic cell a mere relay for reward-prediction bursts generated in the PPN. This of course does not explain how the brain can do reward prediction, it merely passes the buck to the PPN. How closer does this get us to understanding reward prediction? Can the PPN synthesize reward prediction error?

A very insightful and intriguing set of questions – questions that we don’t have answers to. That said, we can speculate. PPN neurons likely provide information about stimulus salience, actual reward presentation and movement initiation to the SNc. These are all events that have been reported to generate SNc bursts. Recent work has shown that SNc DA neurons transiently fire at the onset of movement (Howe and Dombeck, 2016). Cells within the PPN have been shown to also fire prior to movement, with many cells showing directional preference (Thompson and Felsen, 2013). Glutamatergic cells in the PPN have been particularly tied to the role PPN plays in locomotion, with basal ganglia circuits being implicated as regulators of these cells (Roseberry et al., 2016). What is less likely is that PPN provides a reward prediction error signal, as PPN neurons continue to respond to fully conditioned stimuli, regardless of prediction error (Dormont et al., 1998; Kobayashi and Okada, 2007; Okada et al., 2009). This distinction is made in the revision and underscores the heterogeneity in burst generation and SNc signaling.